# Damage Detection and Evaluation for an In-Service Shield Tunnel Based on the Monitored Increment of Neutral Axis Depth Using Long-Gauge Fiber Bragg Grating Sensors

**DOI:** 10.3390/s19081840

**Published:** 2019-04-18

**Authors:** Sheng Shen, Huaxin Lv, Sheng-Lan Ma

**Affiliations:** 1College of Civil Engineering, Fuzhou University, Fuzhou 350108, China; 2Zhongtian Construction Group Zhejiang Steel Structure Co., Ltd., Hangzhou 310008, China; Lvhuaxin@tom.com; 3Fujian Provincial Key Laboratory of Advanced Technology and Informatization in Civil Engineering, Fujian University of Technology, Fuzhou 350118, China; mashenglan@fjut.edu.cn

**Keywords:** damage detection, damage evaluation, shield tunnel, neutral axis depth, long-gauge Fiber Bragg Grating sensors

## Abstract

It is difficult to detect and evaluate the structural damage in a shield tunnel during operation because many traditional techniques based on the observation of vibrations are limited in daily monitoring in tunnels. Thus, the curvature radius of a static longitudinal settlement curve is used to identify the residual health and safety of an in-service shield tunnel. However, there are still two problems. The curvature radius is suitable for a qualitative judgment rather than a quantitative evaluation for longitudinal damage detection. Moreover, the curvature radius, which is calculated from the measured settlements of three neighboring points, gives an average damage degree in a wide scope only and is difficult to use to identify the damage’s precise location. By means of the analysis of three kinds of longitudinal failure modes in a shield tunnel, this paper proposes: (1) a damage detection method based on the monitored increment of the neutral axis depth; and (2) an index to evaluate longitudinal damage. The index is composed of the residual ratios of the equivalent flexural stiffness (HFM1) and the equivalent shear stiffness (HFM3). The neutral axis position and the proposed damage index can be determined using long-gauge Fiber Bragg Grating sensors. Results from numerical simulations show that the deviation between the HFM1 and the true value residual ratio of the equivalent flexural stiffness is no more than 1.7%. The HFM3 is equal to its true value in the entire damage process. A loading experiment for a scaled-down model of a shield tunnel using long-gauge Fiber Bragg Grating sensors indicated that the errors in the HFM1 were no more than 5.0% in the case of early damage development (HFM1 ≥ 0.5). The maximum error did not exceed 9.0% even under severe damage conditions in the model. Meanwhile, the HFM3 also coincided with its true value in the entire testing process.

## 1. Introduction

Due to its safety, high efficiency in construction, and low impact on the urban environment, many metro tunnels have been built by the shield-driven method to satisfy the increasing demand for public transportation in many metropolises. However, some shield tunnels have suffered abrupt damage with severe consequences in recent years. Research points out that a shield tunnel may gradually deteriorate due to many potential risks [1,2], such as groundwater leakage [3,4], nearby excavation or unloading [5,6], transverse deformation [7], longitudinal settlement [8,9], land subsidence [10], earthquake [11,12], and ground fissures [13,14]. As pointed out by Russell and Gilmore [15], these tunnel accidents may be prevented if an aggressive inspection and maintenance procedure for structural damage is implemented.

As a typical prefabricated structure, a shield tunnel is assembled by a number of concrete segments with both longitudinal and circumferential joints. The joints, which contain a series of steel bolts, play a key role in interconnecting the adjacent segments and transferring loads, including axial force, shear force, and the bending moment. Numerical simulations and loading experiments show that the weakest links in the entire structure are the joints rather than the concrete segments [16,17]. Moreover, the decrease in the degrees of flexural stiffness and shear stiffness caused by an outermost bolt failure is larger than that of tensile rigidity in the longitudinal direction. Therefore, it is theoretically acceptable to consider structural damage due to a joint failure as a reduction in flexural stiffness and shear stiffness in the longitudinal direction.

There are two basic considerations when choosing a proper index to evaluate the degree of structural damage. The index is required to be easy to obtain. Additionally, there needs to be a one-to-one correspondence between the damage and the index to evaluate the damage degree quantitatively. According to the classic theory of structural health monitoring [18], there are two kinds of damage indexes: dynamic indexes and static indexes. Dynamic indexes measure dynamic structural responses, and static indexes measure static structural responses. Distinct from the kinds of dynamic indexes that are used to identify damage to bridges and high-rise buildings, a few dynamic indexes have been proposed to detect damage in shield tunnels, which contain torsional wave speed [19,20], a transmissibility function, and cross correlation analysis [21]. However, the feasibility of these dynamic indexes has only been tested via numerical simulation rather than practical monitoring. In fact, even if a shield tunnel can be forced to vibrate artificially, the kinetic energy of the tunnel will dissipate rapidly due to the damping produced by the restrictions from the surrounding soil. This means that, when a large impact force is applied to the inner surface of the tunnel, only a small part of the tunnel near the impact point will briefly vibrate. Therefore, the fact that the vibrations in the shield tunnel are difficult to measure precisely in practice leads to the result that dynamic damage indexes are inapplicable in damage detection for shield tunnels.

In contrast, the curvature radius of the settlement curve is viewed as a static index to identify damage to a shield tunnel. For example, the tunnel owner/operator places a restriction of 15,000 m on the curvature radius of the settlement curve in Shanghai [22]. Although the curvature radius can be easily calculated by the measured settlements of three neighboring points, it also has some drawbacks for damage evaluation from the perspective of the theoretical relationship between the curvature radius and the longitudinal flexural stiffness. First, the curvature radius in a shield tunnel can show qualitatively whether the tunnel structure is damaged or not. It cannot be used to evaluate the damage degree directly. Moreover, the curvature radius is determined theoretically by the interaction between the bending moment and flexural stiffness. After the outermost bolt yields, either the increment in the bending moment or the decrement in flexural stiffness could result in a decrease in the curvature radius. That is, the mentioned one-to-one correspondence cannot be derived by the curvature radius only. Finally, the curvature radius shows an average degree of damage in a wide scope only and is difficult to use to detect the damage’s precise location. This may also bring about difficulties in subsequent maintenance or reinforcement.

Recently, studies have focused on the neutral axis position due to its ability to be sensitive to any structural damage [23]. The neutral axis depth (NAD) is an inherent structural characteristic similar to the geometrical dimension or the cross-sectional moment of inertia. Ni [24] reported that the NAD, as estimated by a Kalman filter, is sensitive to local cracks on a deck section. Sigurdardottir and Glisic [23] pointed out that the NAD can indicate a change in the position of the centroid of stiffness for any load-carrying beam structure. This conclusion was verified by monitoring in the Streicker Bridge at Princeton University and the US202/NJ23 highway overpass in Wayne, New Jersey [23,25]. The modal macro strain (MMS)-based damage index combined with NAD monitoring was used to identify local damage in a similar test in a steel girder bridge [26]. Soman et al. [27] used the NAD to detect different crack sizes, locations, and orientations under different operational loading conditions. In a word, the NAD has been proven to be suitable for damage monitoring in a beam-like structure. Because a shield tunnel can be viewed as approximate to a beam-like structure, it is possible for the NAD to be a useful tool in damage detection and evaluation in shield tunnels.

Another challenge is to obtain the strain distribution across the joint accurately. First, the sensors’ installation and connection is difficult in practice due to the fact that the required quantity of strain sensors may be quite large, as they need to cover a series of circumferential joints in the longitudinal direction. Second, the length of the joint’s area of influence is about 0.2~0.4 m, which is much longer than the length of traditional strain sensors, such as the electrical resistance strain gauge. For the first problem, the fiber Bragg grating (FBG) sensor is a suitable solution because several FBG sensors can be connected by a fiber-optic cable easily. This special characteristic of FBG sensors can not only remove difficulties in sensor installation and reduce the maintenance cost but also simplify the construction of the sensing system. Recently, a multi-FBG sensor monitoring system has been used in several kinds of infrastructure monitoring, such as vibrations in a long-span bridge [28,29,30], scouring in a bridge [31,32], and corrosion and cracking in concrete [33,34,35,36]. The use of similar FBG-based monitoring systems in tunnels has also been reported [37,38,39]. For the second problem, Li [40] has proposed a long-gauge fiber Bragg grating (LFBG) strain sensor that can obtain the average strain of a distance from 0.1 m to 10 m. The sensitivity-improved LFBG sensor [41] has also been developed to measure slight increments in strain. So far, the LFBG sensor has been applied in strain measurement [42], displacement monitoring [43], and measuring neutral axis movement [44].

This paper is organized as follows. Section 2 describes a damage index *D* to define the degree of longitudinal damage in a shield tunnel based on three kinds of longitudinal failure modes. A method for the determination of the NAD based on LFBG sensors is proposed in Section 3 for longitudinal damage detection in shield tunnels. Section 3 also derives the relationships between the damage index *D* and the NAD for damage evaluation. A numerical simulation in Section 4 and a loading test using LFBG sensors for a scaled-down shield tunnel model in Section 5 are carried out to verify the accuracy of the proposed methods for damage detection and evaluation in shield tunnels.

## 2. Damage Index for a Shield Tunnel in the Longitudinal Direction

### 2.1. Failure Modes Based on Two Longitudinal Deformation Modes

A shield tunnel in the longitudinal direction is typically simplified using two models, namely the beam-spring model [45] and the longitudinal continuous model [46]. In the beam-spring model, the segmental ring is viewed as a set of short beams, and the circumferential joint is replaced by axial, shear, and rotational spring elements. In the longitudinal continuous model, the tunnel is simplified to be a homogenous beam with reduced stiffness. The longitudinal continuous model has been widely applied to analyze the internal forces and deformation in shield tunnels based on the Euler–Bernoulli beam theory [47,48]. The homogenous continuous model is more practical to use in a theoretical analysis than the beam-spring model because the values of rigidities of springs in a joint that are required for the latter model are usually hard to determine.

The deformation of a shield tunnel in the longitudinal direction can be divided into two separate modes as shown in Figure 1 [9,49], and this can be done on the basis of the practically measured settlement data from long-term monitoring [8,9]. One mode is the bending mode, where the compression of the concrete segments occurs on one side, and the tension on the other side causes the joints to open and the bolts there to experience tension. The other mode of deformation is the dislocation mode. In this mode, the accumulation of dislocation between adjacent rings leads to differential settlement, and the circumferential joint mainly experiences a shear force. In the dislocation mode, uniform shear strain appears on the entire section of the joint. According to the mentioned analyses, Wu [50] proposed a Timoshenko beam simplified model (TBSM) to simplify the tunnel as a continuous Timoshenko beam with equivalent flexure stiffness (EI)eq and shear stiffness (κGA)eq. The TBSM can describe the dislocation of a shield tunnel in theory; so, this model is used as the basis for the subsequent derivation of the proposed damage index in this paper.

Consideration of the two deformation modes illustrated in Figure 1 leads to the reasonable assumption of three different kinds of failure mode, denoted FM1, FM2, and FM3. FM1 is the bending failure mode, where yielding occurs from the outermost bolt to the inner ones successively in the tensile region before concrete crushing occurs in the compressive region. FM2 is the shearing failure mode, where all of the bolts in the circumferential joint rupture abruptly due to a shearing action. FM3 represents the decrease in equivalent shear stiffness as a result of bolt failure caused by an excessive bending moment applied to the joint. The failures in FM1 and FM3 are ductile while the failure in FM2 is brittle. Thus, the failure of steel bolts in FM1 and FM3 occurs over a relatively long time, and the rupture of steel bolts in FM2 is abrupt.

### 2.2. Damage Index Derivation

The first step in structural damage identification is to divide the entire structure into several independent, similar elements. An element comprises two adjacent half rings and the circumferential joint between them as shown in the range of *l_s_* in Figure 2. This element can be further divided into two parts: the area influenced by the joint, and the uninfluenced area [51]. The area influenced by the joint has a tensile force that is resisted by the bolts, and the compressive force is withstood by the concrete segment. In the uninfluenced area, both the tensile force and the compressive force are borne by the concrete segment.

The degree of residual fitness of a damaged shield tunnel is defined by the residual ratios of *H*_FM1_–*H*_FM3_ that correspond to FM1–FM3, respectively. The superscripts *ud* and *d* denote that the variables used are in the undamaged state or the damaged state, respectively.

According to the TBSM, *H*_FM1_ is defined as follows:(1)$$H_{\mathrm{FM}1}=\frac{{\left(EI\right)}_{eq}^{d}}{{\left(EI\right)}_{eq}^{ud}}=\frac{\eta^{d}EI}{\eta^{ud}EI}=\frac{\eta^{d}}{\eta^{ud}}$$
where *EI* is the flexural stiffness in the longitudinal direction and *η* is the reduction factor of the flexural stiffness. The value of *η^d^* ranges from 0.1 to 0.2 depending on the form of the segment assembly [51].

The value of *H*_FM2_ is expressed as:(2)$$H_{\mathrm{FM}2}=\;0\;or\;1$$
Note that *H*_FM2_ is 0 when the circumferential joint is completely damaged by the shearing action, while an *H*_FM2_ value of 1 means that the joint is undamaged. The discontinuous nature of Equation (2) reflects that damage to the circumferential joint occurs suddenly and that FM2 is brittle.

The expression for *H*_FM3_ is as follows:(3)$$H_{\mathrm{FM}3}=\frac{{\left(\kappa GA\right)}_{\mathrm{eq}}^{d}}{{\left(\kappa GA\right)}_{eq}^{ud}}=\frac{l_{s}}{\frac{l_{b}}{n^{d}\kappa_{b}G_{b}A_{b}}+\frac{l_{s}-l_{b}}{\kappa_{s}G_{s}A_{s}}}/\frac{l_{s}}{\frac{l_{b}}{n^{ud}\kappa_{b}G_{b}A_{b}}+\frac{l_{s}-l_{b}}{\kappa_{s}G_{s}A_{s}}}=\frac{1+\alpha /n^{ud}}{1+\alpha /n^{d}}$$
where *G_b_* is the shear modulus of a bolt, *G_s_* is the shear modulus of the concrete, *A_b_* is the cross-sectional area of the bolt, *A_s_* is the cross-sectional area of the segment, *l_b_* is the length of a bolt, *l_s_* is the length of the segmental ring, *n^ud^* is the number of bolts in the circumferential joint, *n^d^* is the number of un-yielded bolts in the circumferential joint, *κ_b_* is the Timoshenko shear coefficient of the bolt and is 0.9 for a circular cross-section [50], and *κ_s_* is the Timoshenko shear coefficient of the segmental ring and is 0.5 for an annular cross-section [50]. The expression of *α* is as follows:(4)$$\alpha =\frac{l_{b}}{l_{s}-l_{b}}\cdot \frac{\kappa_{s}G_{s}A_{s}}{\kappa_{b}G_{b}A_{b}}=\frac{l_{b}}{l_{s}-l_{b}}\cdot \frac{\kappa_{s}A_{s}E_{s}}{\kappa_{b}A_{b}E_{b}}\cdot \frac{1+\mu_{b}}{1+\mu_{s}}$$
where *E_b_* is the elastic modulus of the steel bolt, *E_s_* is the elastic modulus of the concrete segment, *μ_b_* is the Poisson’s ratio of the steel bolt, and *μ_s_* is the Poisson’s ratio of the concrete segment. It is noted that *α* is a constant for a given tunnel.

Generally, *E_s_*/*E_b_* ≈ 0.15–0.2, *κ_s_*/*κ_b_* ≈ 0.56, *l_b_*/(*l_s_* − *l_b_*) ≈ 0.85–1, (1 + *μ_b_*)/(1 + *μ_s_*) ≈ 1, 10 ≤ *n^ud^* ≤16, and *A_s_*/*A_b_* ≥ 1200. Therefore, *α* is approximately 90–130, and it is much larger than *n^ud^*.

The following simplified approximation of Equation (3) can be used when the structural damage is not severe:(5)$$H_{\mathrm{FM}3}=\frac{1/\alpha +1/n^{ud}}{1/\alpha +1/n^{d}}\approx \frac{n^{d}}{n^{ud}}$$

The damage index *D* is defined as a combination of *H*_FM1_–*H*_FM3_, according to:(6)$$D=1-H_{\mathrm{FM}1}H_{\mathrm{FM}2}H_{\mathrm{FM}3}$$

The probability of FM2 occurring is generally small, so *H*_FM2_ is usually equal to 1. Substituting Equation (1) and Equation (5) into Equation (6) with the assumption that *H*_FM2_ is equal to 1 gives the simplified expression of *D* as follows:(7)$$D\approx 1-H_{\mathrm{FM}1}H_{\mathrm{FM}3}\approx 1-\frac{\eta^{d}}{\eta^{ud}}\frac{n^{d}}{n^{ud}}$$

Two main conclusions can be drawn from the damage index *D* derived from Equations (1)–(7). The first conclusion is that *D* is mainly related to the state of the bolts and is not affected by the concrete segment. Thus, the primary goals when monitoring a shield tunnel should be to observe whether the bolts have yielded, and, if any bolts have yielded, to determine the number of yielded bolts. The second conclusion is that the positions of the neutral axis and yielding line relative to the cross-section of the circumferential joint are important to ascertain. The former is used for the calculation of *η^d^*/*η^ud^*, and the latter is applied to determine *n^d^*/*n^ud^*.

## 3. Damage Detection and Evaluation Using LFBG-Based Strain Measurements

The derivation process in this section is based on the following assumptions: (i) the strain distributions on the circumferential joint and concrete segment are approximately the same as those in the plane-section; (ii) the tensile force in the joint area is resisted entirely by the bolts, and the compressive force in the same area is withstood by the concrete segment; (iii) the pretension force in the bolts and the axial force in the longitudinal direction of the shield tunnel are not considered; (iv) the second failure mode (FM2) is not taken in account; and (v) the bolts are distributed uniformly in the circumferential joint and all bolts have the same mechanical and geometrical properties.

### 3.1. Relationship between Structural Damage Level and NAD

The neutral axis is a horizontal line within each cross-section of a beam where the normal stress and strain is zero. Figure 3 shows that the maximum tensile stress $\sigma_{t}^{ud}$ and the maximum compressive stress $\sigma_{c}^{ud}$ on the cross-section of a rectangular beam are less than the yield stress *σ_y_* in the elastic stage. The NAD *h^ud^* is constant while the bending moment applied to the cross-section increases. However, when the cross-section is in the plastic stage, $\sigma_{t}^{d}$ reaches *σ_y_*, the NAD increases, and *h^d^* is greater than *h^ud^*. Thus, the NAD is sensitive to structural damage. The neutral axis moves towards the compressive region during the development of structural damage. Therefore, the damage level and NAD correspond in beam-like structures, and the damage level can be determined quantitatively by measuring the NAD in practice.

The NAD of the area influenced by the joint has been shown to be distinct from that of the uninfluenced area of the element [46,50]. However, the tensile strain is concentrated mainly at the circumferential joint rather than the segments, so it is suitable to choose the area that is influenced by the joint as the monitoring area for the NAD rather than the uninfluenced area. The influencing factor *λ* of the joints ranges from 0.45 to 0.65 for different forms of the segment assembly [51]. Consequently, if the *l_b_* is assumed to be about 60 cm, the length of the area influenced by the joint is greater than 20–30 cm and exceeds the gauge length of an electrical resistance strain gauge. A solution to this problem is to replace the electrical resistance strain gauge by an LFBG sensor.

### 3.2. Introduction to LFBG Sensors

Figure 4 [40] illustrates the structure of an LFBG sensor, the gauge length of which ranges from several centimeters to several meters. The most remarkable feature of this sensor is that the FBG is sleeved and fixed at both ends of a hollow tube that is protected by the surrounding fiber-reinforced plastic (FRP) package. Thus, the sensing length can be predetermined according to different requirements. Figure 5 shows a comparison of the strain measurement from a traditional short-gauge strain sensor and the LFBG sensor. The short-gauge stain sensor may be ruptured due to the large tensile strain caused by the joint opening. The LFBG sensor can measure the strain safely after the joint opens because the sensor has point fixation only at the two ends bonded to the concrete surface. Thus, the LFBG sensor measures a uniform strain distribution within its gauge length and reduces the risk of rupture when there is excessive joint opening.

### 3.3. Damage Evaluation Using LFBG-Based Strain Measurements

The NAD, and therefore damage, can be determined using an array of LFBG sensors. A series of LFBG sensors, labeled as S1–S*n*, can be fixed on the inner surface of a tunnel as shown in Figure 6a,b. Figure 6c shows that the method of least squares can be used to determine the best-fit of the straight line *y* = *kε* + *c* based on the height coordinate of S1–S*n* and the corresponding strain measurements *ε*_1_–*ε_n_*. The parameter *c* in the fitted equation is equal to the NAD and is denoted as *χ*. The value *ε_y_* is defined as the yield strain of the bolts. If *ε* is substituted by *ε_y_* in the best-fit equation, the calculated *y* is the height coordinate *γ* of the yield line as shown in Figure 6c. It means that the bolts above the yield line have not yielded, but the bolts with positions lower than the yield line have yielded. Thus, the parameter *n^d^* can be obtained easily. In addition, it may be required that a few LFBG sensors, sometimes associated with some other distributed sensing technique [52,53], are placed in one monitoring position in practice to not only ensure accurate measurements for a long time but also eliminate the effect of some unpredicted factors, such as the shrinkage of concrete on the fiber and local compression on the Bragg sensor.

The neutral axis is a horizontal line, so the central angle *φ* corresponding to the neutral axis is:(8)φ=arcsin[2χ/(R+r)]
where *R* and *r* are the outer radius and inner radius of the segment ring, respectively.

Similarly, the central angle *ω* corresponding to the yield line is obtained as follows:(9)ω=arcsin[2γ/(R+r)]

These measurements allow for the determination of the important parameters used in Equation (7). First, the undamaged state is considered. According to the TBSM, *η^ud^* is calculated by Xu [51]: (10)$$\eta^{ud}=\frac{{\left(EI\right)}_{eq}^{ud}}{E_{s}I_{s}}=\frac{\zeta^{ud}l_{s}}{\zeta^{ud}(l_{s}-\lambda l_{b})+\lambda l_{b}}$$
where *E_s_* is the elastic modulus of the concrete segment, and *I_s_* is the area moment of inertia of the cross-section of the concrete segment ring. The coefficient of rotational stiffness *ζ^ud^* for a circumferential joint is determined from Shiba [46]:(11)$$\zeta^{ud}=\frac{\cos^{3}\varphi^{ud}}{\cos\varphi^{ud}+\left(\varphi^{ud}+\pi /2\right)\cdot \sin\varphi^{ud}}$$

The following theoretical method for the calculation of *φ^ud^* was proposed by Xu [51]:(12)$$\cot\varphi^{ud}+\varphi^{ud}=\pi \left(\frac{1}{2}+\frac{K_{j1}\cdot l_{b}}{E_{s}A_{s}}\right)$$
where *K_j_*_1_ = *n^ud^k_j_*_1_ and *k_j_*_1_ is the translational stiffness of the bolt at the joint. The value *φ^ud^* remains constant in the longitudinal direction of a non-destructive shield tunnel. The calculated *φ^ud^* is used to determine *ζ^ud^* and *η^ud^*. All of these values are independent of the strain measurements.

Next, damage evaluation using the long-gauge strain measurements is examined. Shiba [46] proposed the expression of (*EI*${)}_{eq}^{d}$ as follows:(13)$${\left(EI\right)}_{eq}^{d}=\frac{2M}{N_{y}r}R_{1}(\sin\varphi -\sin\omega )E_{s}I_{s}$$
where *M* is the moment applied on the segment ring, and *N_y_* is the ultimate axial force of the segment ring. The parameter *R*_1_ in Equation (13) is given by:(14)$$R_{1}=1/\left(1+\frac{E_{s}A_{s}}{l_{s}K_{j1}}\right)$$

Therefore, *η^d^* can be obtained from:(15)$$\eta^{d}=\frac{{\left(EI\right)}_{eq}^{d}}{E_{s}I_{s}}=\frac{2M}{N_{y}r}R_{1}(\sin\varphi -\sin\omega )$$

The equilibrium equation and compatibility equation at a joint are solved simultaneously, and the solution was determined by Shiba [46] to be:(16)$$\frac{2\pi MR_{1}}{N_{y}r}(\sin\varphi -\sin\omega )-\frac{\pi }{2}(1+R_{2})-(1-R_{1})(\varphi +\sin\varphi \cos\varphi )-(R_{1}-R_{2})(\omega +\sin\omega \cos\omega )=0$$
where the parameter *R*_2_ is calculated by: (17)$$R_{2}=1/\left(1+\frac{E_{s}A_{s}}{l_{s}K_{j2}}\right)$$
where *K_j_*_2_ = *n^ud^k_j_*_2_. *k_j_*_2_ is the plastic stiffness of the bolts at a joint.

Substituting Equation (15) into Equation (16), the following expression of *η^d^* is obtained:(18)$$\eta^{d}=\frac{1}{2}(1+R_{2})+\frac{1}{\pi }[(1-R_{1})(\varphi +\frac{1}{2}\sin2\varphi )+(R_{1}-R_{2})(\omega +\frac{1}{2}\sin2\omega )]$$

The value of *η^d^* in Equation (18) depends on the *φ* and *ω* as determined from Equation (8) and Equation (9) because *R*_1_ and *R*_2_ are related only to the mechanical and geometrical characteristics of the bolts. Thus, *η^d^* is determined by the long-gauge strain measurements shown in Figure 5.

The calculated *η^ud^*, *η^d^*, *n^ud^*, and *n^d^* are substituted into Equation (7) to determine the value of the damage index *D*.

## 4. Numerical Simulation Verification

A numerical model was used to simulate the shield tunnel in the longitudinal direction and verify the applicability of the damage index *D* for damage detection and evaluation. The decrease in the longitudinal stiffness is compared to the damage index *D* calculated from the long-gauge strain measurements in the model. Verification using this model is concerned with three main aspects. First, the plane-section assumption should be shown to be applicable before and after damage, and that the NAD is sensitive enough to indicate the damage. Second, the theoretical *φ^ud^* should be close to its true value at the undamaged state. Finally, the monitored damage level from the LFBG sensors should approximate the true damage level.

### 4.1. Numerical Model

The numerical segment ring built in the ANSYS software is based on a real segment ring in the Shanghai Metro Line No. 1. The mechanical properties of the segment and bolt are as follows: *E_s_* = 34.5 GPa, *E_b_* = 206 GPa, *μ_s_* = 0.167, *μ_b_* = 0.2, the compressive strength of the concrete = 32.4 MPa, the tensile strength of the concrete = 1.89 MPa, the yield strength of a bolt = 640 MPa (the yield strain of a bolt = 3107 με, correspondingly), the ultimate strength of a bolt = 800 MPa, and *k_j_*_2_/*k_j_*_1_ = 0.1. The thickness of the segment is 0.3 m, and the width of the ring is 1.5 m. The outside diameter of the ring is 6.0 m, and the inside diameter of the ring is 5.4 m. As shown in Figure 6, the segment ring is composed of one Block K (22.5°) and five standard Block As (67.5°). As shown in Figure 7, there are 20 rings assembled by segments with straight joints. A ring contains 16 circumferential weld bolts (M24) and 12 longitudinal joint bolts (M27). The value of *l_b_* is 0.6 m. The two half-segment rings at both ends are not considered, and the model is uniformly divided into 19 elements denoted E1–E19.

The concrete segments are simulated using the Solid65 element. The shear transfer coefficient is set to 0.5 and 0.9 for crack opening and crack closing, respectively. The bolts are simulated by using the Solid185 element. In this model, the two ends of each bolt are embedded in the corresponding segments. Contact elements are used to simulate the contact and friction between the contacting surfaces. The end surfaces of each bolt are set as the target surface in the TARGE170 contact element, and the surfaces of the bolt holes are set as the contact surface in the CONTA173 contact element. The friction coefficient between the contacting surfaces is set to 0.3. Furthermore, the surface-to-surface contact is simulated between the contacting end surfaces of adjacent segments. One end surface is set as the target surface in the TARGE170 contact element, and the other is set as the contact surface in the CONTA173 contact element. The friction coefficient between the contacting surfaces is set to 0.7. The penetration coefficient in each contact element is set to 1. Further details of the contact settings in this model are given by Zhong [54].

### 4.2. Sensor Placement and Loading Mode

The LFBG sensors, denoted S1–S9, are fixed on the inner surface of the area influenced by the joint in E6 (one-third span) as shown in Figure 8. This figure also gives the height coordinates of S1–S9 by setting the horizontal axis at the center of the cross-section of the ring. The bolts located on the outside of the sensors are named B1–B9. The sensor placement for E9 (mid-span) is the same. These sensors are simulated by the Beam188 element. All sensors have the uniform gauge length of 300 mm and the uniform width of 10 mm. The elastic modulus of the sensors is set to 8.0 MPa. The two ends of the sensors are glued to the inner surface of the segment, and the sensing region is separated from the segment.

Displacement constraints are applied in the *z*-direction at both ends of the model and in the *x*-direction on the outer surface of the ring line, as shown in Figure 9. The uniform load *p* is applied by seven successive loading steps from 0 to 30 kPa, 60 kPa, 90 kPa, 105 kPa, 120 kPa, 150 kPa, and 180 kPa.

### 4.3. Results and Analysis

#### 4.3.1. Verification of Plane-Section Assumption and NAD Sensitivity

Table 1 and Table 2 show the tensile and compressive strain measurements for E9 and E6 at each loading step. The tensile strain is defined to be positive. Figure 10 shows that there are approximately linear relationships between the measured strains and the heights of the sensors. Thus, the method of least squares easily obtains the linear equations of *y* = *kε* + *c* for each linear relationship. These equations are used to construct Table 3 that shows the coefficient of determination *R*^2^ and the parameter *c* that is equal to the monitored NAD *χ* for each equation. The NAD of E9 remains approximately 2.560 m when the applied load *p* is less than 105 kPa. The value of *χ* begins to increase when *p* reaches 105 kPa. For E6, the same trend of *χ* also occurs for *p* greater than 120 kPa. Thus, the proposed detecting method in Section 3 defines *p* = 105 kPa and *p* = 120 kPa as the threshold values of structural damage in E9 and E6, respectively. The stresses on bolts at different loading steps for E9 and E6 are listed in Table 4 and Table 5, respectively. Table 4 shows that the stress of the outermost bolt B9 in the tensile region of E9 reaches 661 MPa for a *p* of 105 kPa. This stress value of this bolt slightly exceeds the yield stress of 640 MPa. Table 5 shows that the stress of B9 in the tensile region of E6 reached 651 MPa at a *p* of 120 kPa. This bolt stress is also larger than the 640 MPa yield stress. These stress values indicate that the NAD *χ* is a sensitive index that can indicate the occurrence of damage. Moreover, the applicability of the plane-section assumption is verified since the coefficient of determination *R*^2^ of each best-fitting straight line is greater than 0.997 for each loading step before and after damage.

#### 4.3.2. Accuracy Comparison of *φ^ud^*

The theoretical *φ^ud^* is obtained from the proposed models of Xu [51] and Shiba [46]. In the latter work, *φ^ud^* is determined from: (19)$$\cot\varphi^{ud}+\varphi^{ud}=\pi \left(\frac{1}{2}+\frac{K_{j1}\cdot l}{E_{s}A_{s}}\right)$$
where the parameter *l* in Equation (19) represents the whole width of the ring. Shiba [46] proposed that the length of the area influenced by the joint should be equal to *l*. However, Xu [51] believed that the length may be less than *l*. Therefore, the parameter *l* in Equation (19) is replaced by *l_b_* as shown previously in Equation (12). The two models produce different *φ^ud^* values, so it is necessary to ascertain which model has better accuracy.

The mechanical and geometrical properties of the real tunnel are used for the comparison of the theoretical *φ^ud^* as calculated by Equation (19), Equation (12), and Equation (8). The results of the comparison are given in Table 6. The value of *φ^ud^* obtained from Xu [51] is closer to the result from the numerical model than that from Equation (19). Thus, Xu’s model is more accurate than Shiba’s model. According to Xu [51], *λ* is 0.6225 for the case of straight-jointed segmental tunnel lining. Substituting this value of *λ* into Equation (10) results in a *η^ud^* of 0.126.

#### 4.3.3. Verification of the Damage Index’s Accuracy

The damage index *D* is determined by *H*_FM1_ and *H*_FM3_ according to Equation (7). The verification for the accuracy of this *D* can be divided into two parts: verifying the accuracy of *H*_FM1_ and verifying the accuracy of *H*_FM3_. The following equation is used to obtain the practical flexural stiffness at each loading step before or after damage:(20)$$EI=\bar{M}l/\bar{\theta }$$
where *M* is the average moment of the monitored element, and *θ* is the average slope of the monitored element. The calculation for this case shows that the theoretical flexural stiffness of the cross-section of the segment is 7.544 × 10^8^ kN·m^2^. Table 7 and Table 8 give the comparisons between the (*EI*)*^d^*/(*EI*)*^ud^* and *H*_FM1_ for E9 and E6, respectively. The *H*_FM1_ value is very close to the value of (*EI*)*^d^*/(*EI*)*^ud^* before and after damage occurrence. The maximum error between the (*EI*)*^d^*/(*EI*)*^ud^* and the *H*_FM1_ in E9 and E6 is no more than 1.7% and 0.8%, respectively.

The calculated *γ* is used for determining *n^d^* at each loading step. Table 9 gives the comparison of this determined *n^d^* in E9 to that established from the fitting equations based on the long-gauge strain measurements and the stress on the bolt shown in Table 4. Table 10 shows the same comparison for E6. It is found that the *n^d^* determined from the fitting equations is equal to the true *n^d^* in the entire damage process.

## 5. Experimental Verification

A scaled-down longitudinal concrete model was employed to simulate an actual shield tunnel. The damage levels are monitored and compared to the true damage levels. This comparison is important to the applicability of the proposed damage detection and evaluation methods for practical monitoring.

### 5.1. Scaled-Down Model

The Qingchun Road of Hangzhou shield tunnel was selected as the prototype for the scaled-down model’s design. The similarity ratio of the geometric dimensions and the elastic modulus in the model is 7 and 1, respectively. The M40 bolts in the real tunnel were simulated by bent screws with a uniform diameter of 6 mm. The normal concrete in the real tunnel was modeled by a fine aggregate concrete. The mechanical properties of the segment and bolt were obtained by laboratory tests and are as follows: *E_s_* = 35 GPa, *E_b_* = 190 GPa, *μ_s_* = 0.2, *μ_b_* = 0.3, the compressive strength of the concrete = 41.4 MPa, the yield stress of a bolt = 710.08 MPa, the yield strain of a bolt = 3680 με, and *k_j_*_2_/*k_j_*_1_ = 0.001. The scaled-down model has 20 segment rings. The outside diameter is 860 mm and the inside diameter is 780 mm. The width and the thickness of a segment ring is 200 mm and 40 mm, respectively. The segment ring is composed of six standard Block As (60°) with staggered joints. The staggered angle between the two adjacent rings is 20°. A joint of the lining ring contains 12 circumferential weld bolts and 18 longitudinal joint bolts. The value of *l_b_* is 0.1 m. Figure 11 shows the components of the model.

### 5.2. Sensor Placement and Loading Mode

The two half-segment rings at both ends were ignored, and the remaining model was uniformly divided into 19 elements denoted E1–E19. Figure 12a shows the placement locations of 10 LFBG sensors with a uniform length of 70 mm that were fixed on the inner surface of the area influenced by the joint in E8. These sensors are denoted S1–S10. The maximum strain that can be measured by these strain sensors is approximately 3600 με. The bolts on the outside of the sensors are labeled B1–B10. As shown in Figure 12b, each bolt of B1–B10 is attached to an electrical resistance strain gauge with a uniform size of 4 mm × 3 mm to measure the tensile or compressive strain in the bolt at each loading step. These sensors have a maximum strain measurement of approximately 4500 με. The sensor placement for E10 is the same as E8.

Two concrete supports with a curved groove were placed at both ends of the model as shown in Figure 13. Two curved steel sheets with a length of 650 mm were laid between the model and the groove to smooth the stress concentration and prevent shear failure. The concentrated load produced by an electro-hydraulic servo loading system was divided equally into two parts by a transfer steel board. Each part of the load passes through a steel plate designed to have the same curved surfaces as the steel sheets. Rubber pads were placed between the plates and the model to ensure a smooth load transfer process. The applied load *P* was increased by using displacement control from a linear variable differential transformer installed in the loading system. The applied displacement *y* of the loading system was increased by 1 mm for each step from *y* = 0 to *y* = 12 mm. The two midpoints of the uniform loads are located at the boundary point between E8 and E9 and the boundary point between E11 and E12. The two supporting points are located at the boundary point between E1 and E2 and the boundary point between E18 and E19. The increase in speed of the applied displacement is 0.5 mm per minute to prevent an accidental rupture that may occur at any circumferential joint. A schematic and the setup of the loading system are shown in Figure 13. The applied load *P* measured at each loading step is listed in Table 11.

### 5.3. Results and Analysis

#### 5.3.1. Verification of Plane-Section Assumption and NAD Sensitivity

Table 12 and Table 13 show the measured strains at different positions of the area influenced by the joint in E10 and E8 at each loading step, respectively. Figure 14 illustrates that linear relationships exist between the strains and the corresponding height coordinates of the sensors. Fitting the linear expression *y* = *kε* + *c* based on the least squares method gives the values of the coefficient of determination *R*^2^, the monitored *χ*, and *γ* listed in Table 14. It is found that *χ* of E10 remains in the range of 321.0 mm to 321.4 mm when *y* is less than 6 mm. Then, *χ* begins to increase when *y* increases beyond 6 mm. The similar point in E8 is a *y* of 7 mm. Table 15 shows that, when the loading step of *y* is 6 mm, the strain of 3615 με found in the outermost bolt B10 of E10 is very close to the yield strain of a bolt. The strain of 3658 με for the same bolt B10 in E8 is close to the yield strain of bolt when *y* is 7 mm as seen in Table 16. Therefore, the sensitivity of the NAD to damage is confirmed by the occurrence of damage based on the coincidence between the increment of *χ* and the strain measurements of the bolts. Also, the *R*^2^ for each fitting line is greater than 0.980 before any damage. This value decreases gradually with damage development. Thus, the decrease in *R*^2^ may also be considered to show the damage development. Furthermore, the values of *R*^2^ given in Table 14 indicate that the plane-section assumption is applicable in the undamaged state and the early-damaged state.

#### 5.3.2. Verification of the Damage Index’s Accuracy

The applied load *P* and the positions of loading points and supporting points are used to obtain the bending moment *M* acting on the test specimen. The curvatures *κ* of the testing specimens at each loading step are obtained by the fitting equations to the monitored long-gauge strains. The practical flexural stiffness of the monitored element before or after damage at each loading step is calculated according to:(21)EI=M/κ
According to Xu [51], *λ* is 0.535 for the case of a stagger-jointed segmental tunnel lining. Substituting a *λ* of 0.535 into Equation (10), the value of *η^ud^* is determined to be 0.144. The comparison between the (*EI*)*^d^*/(*EI*)*^ud^* and the *H*_FM1_ for E10 and E8 from the scaled-down model is shown in Table 17 and Table 18, respectively. It is significant that *H*_FM1_ is equal to (*EI*)*^d^*/(*EI*)*^ud^* in the non-damaged state, and the deviations between the two ratios are no more than 5.0% for the early-damaged state (*H*_FM1_ ≥ 0.5) in both E10 or E8. The maximum error would not exceed 9.0% during the full damage process.

Table 19 gives the comparison of *n^d^* for E10 from the fitting equations based on monitored long-gauge strains and from those listed in Table 14. Table 20 presents the same comparison of *n^d^* for E8. The results show that the *n^d^* determined from the fitting equations is equal to the *n^d^* as monitored from strains at each loading step. Thus, the proposed *H*_FM3_ in this paper is an accurate damage index in both the undamaged state and the damaged state.

## 6. Conclusions

This study proposes a new damaged detection and evaluation method based on NAD monitoring in the longitudinal direction for in-service shield tunnels. Verifications of the proposed method were performed using numerical simulations and experiments, and the following conclusions can be drawn:Based on the analysis of two deformation modes (i.e., bending mode and dislocation mode), three kinds of failure modes are assumed, including the bending failure mode, the shearing failure mode due to dislocation, and the shearing failure mode of the equivalent shear stiffness reduction, which resulted from the bolts being subjected to an excessive bending moment. These three failure modes and their development were successfully detected and evaluated by the proposed damage index *D*.The damage index *D* can be determined only by the monitored NAD, which is obtained from the long-gauge strain distribution of the LFBG sensor array. The calculation of the damage index value is independent of the loading mode and the mechanical properties of materials.The damage index *D* characterizes the development of damage in the tunnel accurately through calculating the coefficients *H*_FM1_ and *H*_FM3_ in most cases. The maximum deviation between the *H*_FM1_ and the true decrease of the equivalent flexural stiffness is less than 1.7% in the numerical simulation result and less than 9% in the experimental result. The calculated *H*_FM3_ matches well with its true value both in the undamaged state and the early-damaged state in the numerical simulation and the experimental results. The accuracy of the index *D* demonstrates the great applicability of the proposed damaged detection and evaluation method in the practical monitoring and maintenance of in-service shield tunnels.

## Figures and Tables

**Figure 1 sensors-19-01840-f001:**
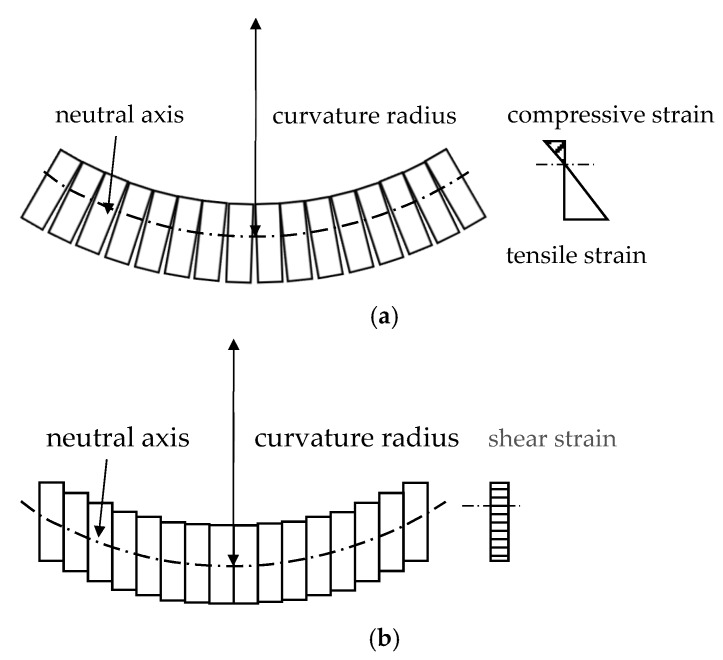
The two longitudinal deformation modes of a shield tunnel. (**a**) Bending mode; (**b**) Dislocation mode.

**Figure 2 sensors-19-01840-f002:**
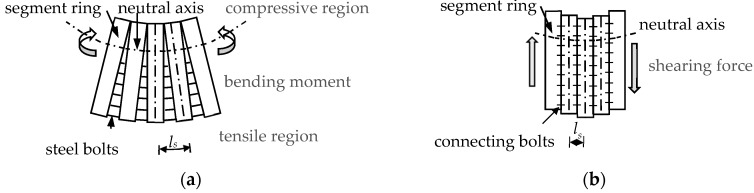
The diagrams used for the definition of structural damage. (**a**) Bending mode; (**b**) Dislocation mode.

**Figure 3 sensors-19-01840-f003:**
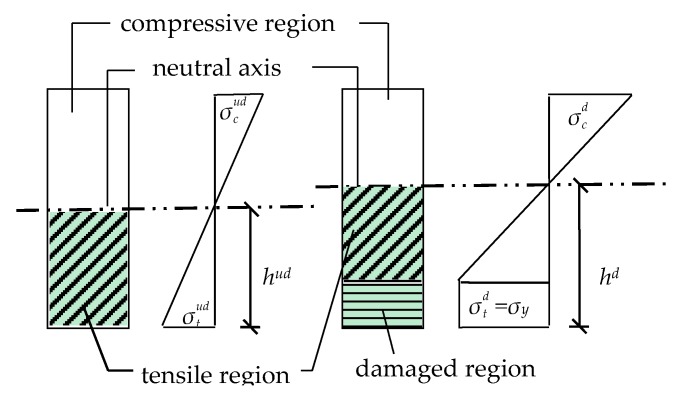
Damage development with the increase of neutral axis depth (NAD) in a rectangular beam.

**Figure 4 sensors-19-01840-f004:**
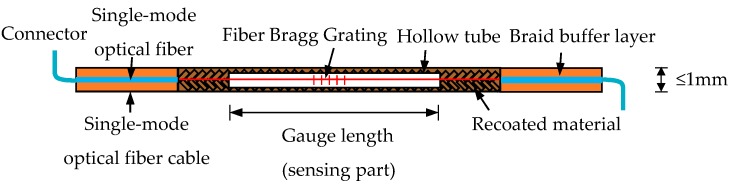
The structural design of the packaged LFBG sensor [40].

**Figure 5 sensors-19-01840-f005:**
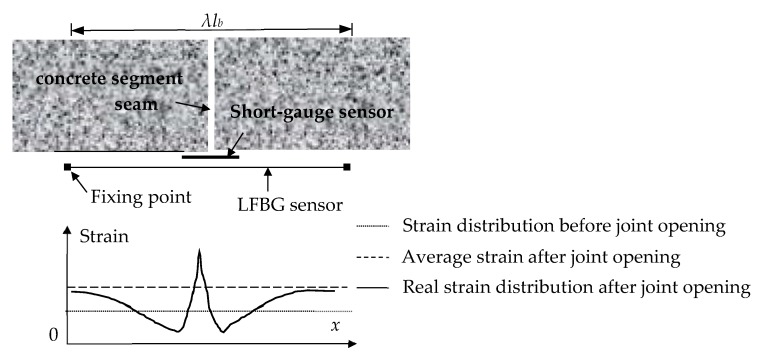
Comparison of strain measurements from the two sensors before and after joint opening.

**Figure 6 sensors-19-01840-f006:**
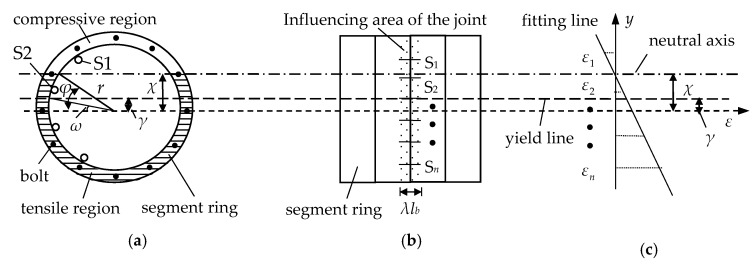
Determination of the NAD in a segment ring based on LFBG strain measurements. (**a**) Circumferential section; (**b**) Longitudinal section; (**c**) Fitting line based on LFBG measurements.

**Figure 7 sensors-19-01840-f007:**
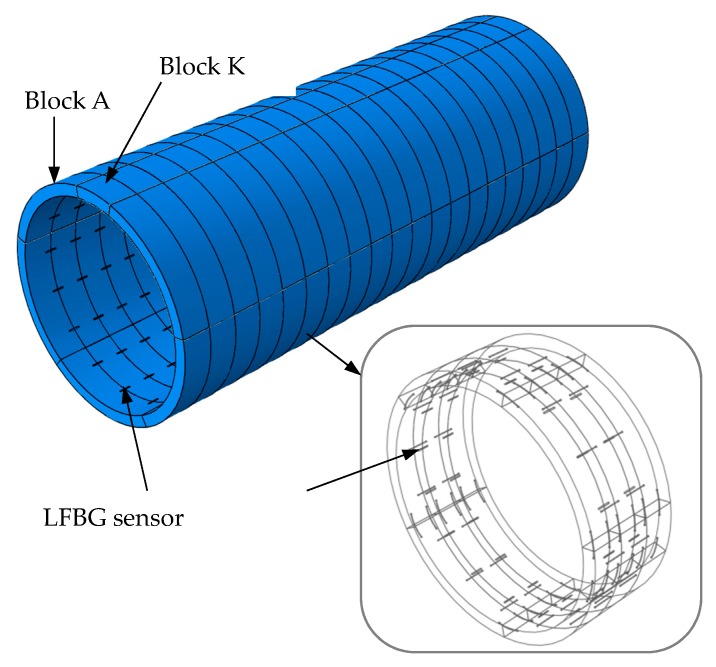
The segment ring model.

**Figure 8 sensors-19-01840-f008:**
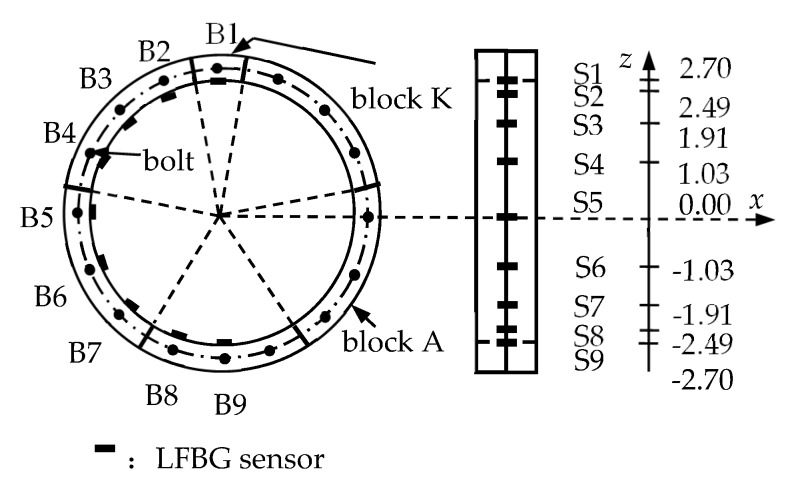
The LFBG sensor placement in one segment (Unit: m).

**Figure 9 sensors-19-01840-f009:**
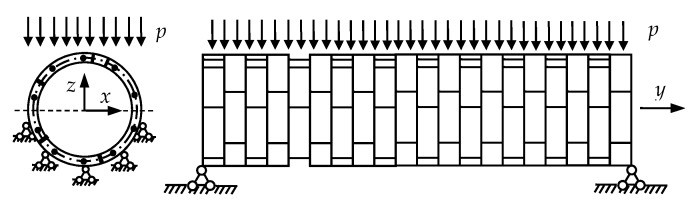
The boundary conditions and uniform load applied to the model.

**Figure 10 sensors-19-01840-f010:**
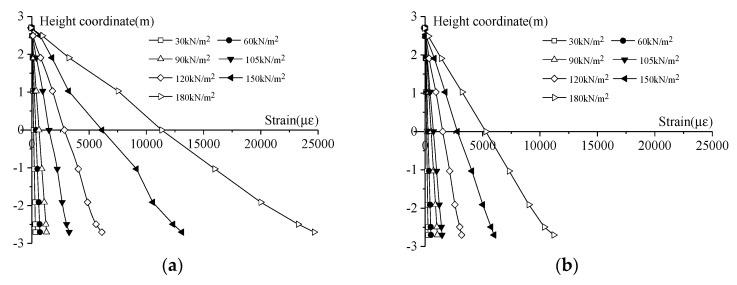
Deformation at different positions in the area influenced by the joint. (**a**) E9; (**b**) E6.

**Figure 11 sensors-19-01840-f011:**
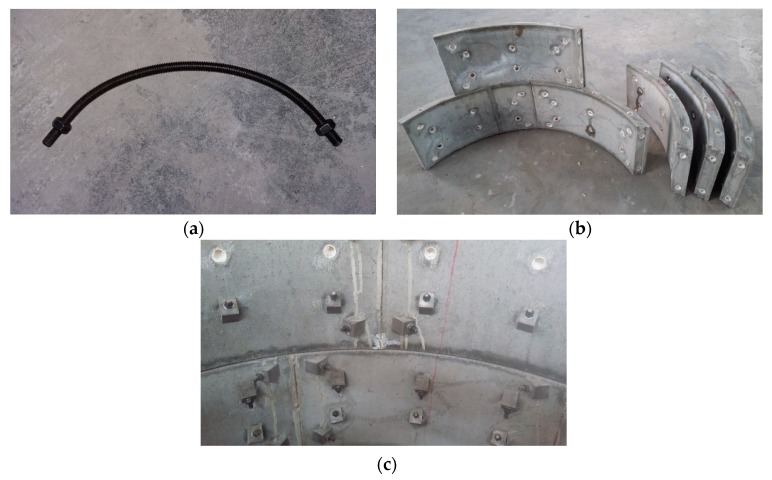
Images showing the components of the scaled-down model. (**a**) Bent screw; (**b**) Concrete segment; (**c**) Longitudinal and circumferential joints.

**Figure 12 sensors-19-01840-f012:**
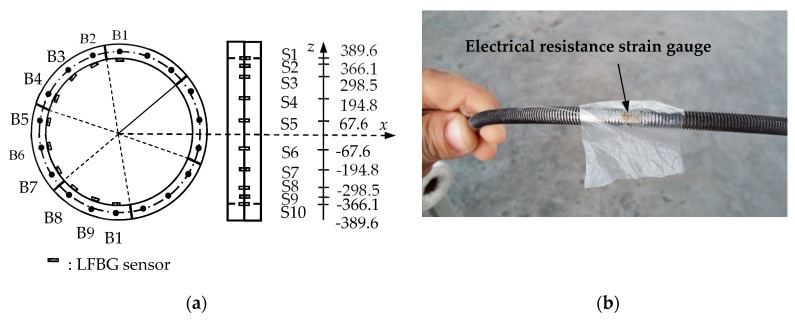
Sensor placement in the scaled-down model. (**a**) S1–S10 (Unit: mm); (**b**) A strain gauge attached to a bolt.

**Figure 13 sensors-19-01840-f013:**
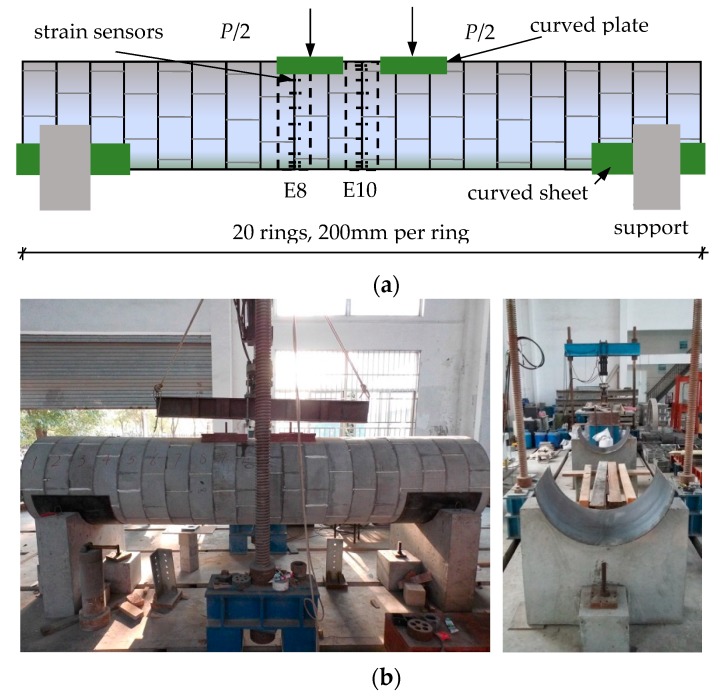
The loading system and the loading process setup. (**a**) Schematic of the loading system; (**b**) Photographs of the loading process and the model support.

**Figure 14 sensors-19-01840-f014:**
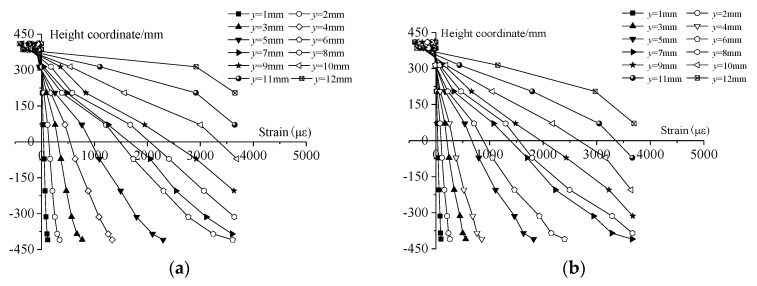
The strain measurements at each loading step in the areas influenced by a joint. (**a**) E10; (**b**) E8.

**Table 1 sensors-19-01840-t001:** The measured strain of each sensor for E9 (unit: με).

Sensor	*p* (kPa)						
	30	60	90	105	120	150	180
S1	−7	−20	−33	−43	−63	−87	−103
S2	3	10	17	57	97	707	900
S3	37	87	157	387	757	1747	3230
S4	87	203	370	947	1793	3217	7523
S5	147	340	620	1497	2833	6130	11,333
S6	207	477	867	2213	4057	9133	15,977
S7	257	593	1080	2640	4857	10,567	19,997
S8	287	670	1220	3033	5600	12,333	23,263
S9	303	700	1270	3253	6110	13,107	24,657

**Table 2 sensors-19-01840-t002:** The measured strain of each sensor for E6 (unit: με).

Sensor	*p* (kPa)						
	30	60	90	105	120	150	180
S1	−7	−13	−27	−40	−47	−60	−67
S2	3	7	13	20	57	70	270
S3	30	63	130	183	347	800	1400
S4	73	147	310	427	933	1743	3210
S5	120	243	517	713	1523	2803	5277
S6	170	340	723	1000	2133	4067	7307
S7	213	423	903	1227	2590	5013	9033
S8	240	480	1017	1413	3010	5733	10,367
S9	250	500	1063	1467	3170	5993	11,197

**Table 3 sensors-19-01840-t003:** The values of *χ*, *γ*, and *R*^2^ from the linear best-fitting equation at each loading step.

Element		*p* (kPa)						
		30	60	90	105	120	150	180
E9	*χ* (m)	2.558	2.558	2.561	2.576	2.593	2.616	2.640
	*γ* (m)	<−2.7	<−2.7	<−2.7	−2.71	−0.20	1.37	2.02
	*R* ^2^	1.000	1.000	1.000	0.999	0.998	0.997	0.998
E6	*χ* (m)	2.558	2.560	2.561	2.561	2.571	2.586	2.607
	*γ* (m)	<−2.7	<−2.7	<−2.7	<−2.7	−2.71	−0.21	1.10
	*R* ^2^	1.000	1.000	1.000	0.999	0.999	0.999	0.999

**Table 4 sensors-19-01840-t004:** The stress of bolts at each loading step in E9 (unit: MPa).

Bolt	*p* (kPa)						
	30	60	90	105	120	150	180
B1	−1	−2	−5	−6	−7	−10	−18
B2	1	4	9	10	71	136	113
B3	8	25	54	76	207	297	671
B4	20	63	136	197	347	666	701
B5	33	103	221	326	524	690	719
B6	45	139	299	452	679	703	732
B7	56	176	374	539	691	719	756
B8	65	195	427	619	704	729	776
B9	68	204	447	661	712	736	788

**Table 5 sensors-19-01840-t005:** The stress of bolts at each loading step in E6 (unit: MPa).

Bolt	*p* (kPa)						
	30	60	90	105	120	150	180
B1	−1	−1	−4	−5	−6	−8	−10
B2	1	2	6	5	10	39	102
B3	6	12	37	61	95	223	611
B4	16	31	94	164	234	513	642
B5	26	51	154	286	390	618	692
B6	35	69	208	401	514	671	711
B7	44	88	263	505	600	701	731
B8	50	101	301	591	639	713	754
B9	53	106	318	621	651	724	762

**Table 6 sensors-19-01840-t006:** Comparison of *φ^ud^* from different models.

	Equation (19) (Shiba [46])	Equation (12) (Xu [51])	Equation (Numerical Model)
*φ^ud^*/°	53.9	62.7	63.9

**Table 7 sensors-19-01840-t007:** Comparison between (*EI*)*^d^*/(*EI*)*^ud^* and *H*_FM1_ in E9.

		*p* (kPa)						
		30	60	90	105	120	150	180
**Numerical model**	*M* (10^4^ kN·m)	2.005	4.010	6.014	7.351	9.356	11.360	13.365
	*θ* (10^−4^)	1.974	3.949	5.923	8.676	16.382	36.411	130.179
	*EI* (10^8^ kN·m^2^)	1.523	1.523	1.523	1.116	0.857	0.468	0.154
	(*EI*)*^d^*/(*EI*)*^ud^*	1.000	1.000	1.000	0.617	0.503	0.323	0.117
**Monitoring**	*η^ud^*	0.126	0.126	0.126	0.126	0.126	0.126	0.126
	*η^d^*	0.126	0.126	0.126	0.078	0.064	0.041	0.015
	*H* _FM1_	1.000	1.000	1.000	0.619	0.508	0.325	0.119
	Error (%)	0.0	0.0	0.0	0.3	1.0	0.6	1.7

**Table 8 sensors-19-01840-t008:** Comparison between (*EI*)*^d^*/(*EI*)*^ud^* and *H*_FM1_ in E6.

		*p* (kPa)						
		30	60	90	105	120	150	180
**From Numerical Model**	*M* (10^4^ kN·m)	1.800	3.600	5.400	6.300	7.200	9.000	10.800
	*θ* (10^−4^)	1.773	3.546	5.318	6.500	8.247	20.265	44.118
	*EI* (10^8^ kN·m^2^)	1.523	1.523	1.523	1.523	1.172	0.755	0.408
	(*EI*)*^d^*/(*EI*)*^ud^*	1.000	1.000	1.000	1.000	0.770	0.496	0.268
**From Monitoring**	*η^ud^*	0.126	0.126	0.126	0.126	0.126	0.126	0.126
	*η^d^*	0.126	0.126	0.126	0.126	0.097	0.063	0.034
	*H* _FM1_	1.000	1.000	1.000	1.000	0.770	0.500	0.270
	Error (%)	0.0	0.0	0.0	0.0	0.0	0.8	0.7

**Table 9 sensors-19-01840-t009:** Comparison of *n^d^* for E9.

		*p* (kPa)						
		30	60	90	105	120	150	180
**Table 4**	*n^d^*	16	16	16	15	9	5	3
**From Monitoring**	*n^d^*	16	16	16	15	9	5	3

**Table 10 sensors-19-01840-t010:** Comparison of *n^d^* for E6.

		*p* (kPa)						
		30	60	90	105	120	150	180
**Table 5**	*n^d^*	16	16	16	16	15	9	5
**From Monitoring**	*n^d^*	16	16	16	16	15	9	5

**Table 11 sensors-19-01840-t011:** The applied load *P* at each loading step.

***y* (mm)**	1	2	3	4	5	6	7	8	9	10	11	12
***P*/kN**	0.286	1.499	3.071	4.57	6.066	6.784	7.354	7.999	8.571	9.144	9.786	10.071

**Table 12 sensors-19-01840-t012:** The strain measurements in the area influenced by the joint in E10 (unit: mm).

Sensor	*y* (mm)											
	1	2	3	4	5	6	7	8	9	10	11	12
S1	−3	−8	−45	−76	−131	−175	−190	−244	−260	−370	−390	−420
S2	−1	−7	−33	−64	−90	−110	−95	−60	−20	1	60	690
S3	4	5	8	23	44	63	77	100	155	629	1602	3120
S4	18	50	105	291	479	825	979	1182	1558	2067	3256	3555
S5	30	80	317	492	845	1315	1559	1877	2146	3302	3649	− *
S6	49	110	452	663	1142	1695	1752	2006	3420	3692	−	−
S7	70	160	520	850	1507	2095	2339	3468	3536	−	−	−
S8	88	210	603	1080	1901	3314	3412	3544	−	−	−	−
S9	104	236	709	1261	2150	3473	3659	−	−	−	−	−
S10	114	270	854	1495	2624	3578	−	−	−	−	−	−

* The sensor is ruptured by strain overloading.

**Table 13 sensors-19-01840-t013:** The strain measurements in the area influenced by the joint in E8 (unit: mm).

Sensor	*y* (mm)											
	1	2	3	4	5	6	7	8	9	10	11	12
S1	−3	−7	−42	−71	−122	−150	−173	−188	−226	−241	−385	−398
S2	−1	−6	−31	−59	−84	−103	−109	−94	−56	−19	2	57
S3	4	5	7	21	41	50	62	76	93	144	590	1555
S4	17	46	97	270	446	545	817	969	1245	1504	2296	3273
S5	28	74	294	457	784	964	1302	1543	1812	2071	3346	3698
S6	45	105	419	615	1060	1304	1678	1814	1936	3405	3666	−
S7	65	148	483	789	1398	1720	2074	2316	3347	3516	−	−
S8	82	195	570	1005	1764	2170	3361	3478	3525	−	−	−
S9	97	226	650	1170	1995	2441	3438	3622	−	−	−	−
S10	106	252	793	1387	2435	2995	3542	−	−	−	−	−

**Table 14 sensors-19-01840-t014:** The values of *χ*, *γ*, and *R*^2^ from the linear best-fitting equation for each loading step.

Element		*y* (mm)											
		1	2	3	4	5	6	7	8	9	10	11	12
**E10**	*χ* (mm)	321.4	321.1	321.0	321.0	321.0	327.3	338.5	347.1	356.9	367.1	377.2	384.4
	*γ* (mm)	<389.6	<389.6	<389.6	<389.6	<389.6	389.0	−354.5	−251.7	−125.5	7.2	132.0	238.1
	*R* ^2^	0.981	0.980	0.980	0.980	0.980	0.979	0.977	0.970	0.965	0.960	0.910	0.829
E8	*χ* (mm)	321.2	321.2	321.0	321.0	321.0	320.9	326.2	338.7	348.7	355.8	366.4	376.5
	*γ* (mm)	<389.6	<389.6	<389.6	<389.6	<389.6	<389.6	−389.6	−352.2	−264.1	−131.1	15.3	133.4
	*R* ^2^	0.981	0.980	0.981	0.980	0.980	0.980	0.978	0.976	0.968	0.967	0.941	0.916

**Table 15 sensors-19-01840-t015:** The strain on the bolts at each loading step in E10 (unit: με).

Bolt	*y* (mm)											
	1	2	3	4	5	6	7	8	9	10	11	12
B1	−11	−37	−84	−116	−153	−175	−186	−279	−322	−474	−541	−597
B2	−8	−27	−61	−76	−183	−129	−100	−70	−52	−31	−11	30
B3	5	17	38	55	84	97	170	302	328	436	711	1273
B4	12	39	88	153	277	381	598	801	1179	1260	2131	3537
B5	35	118	267	394	705	1061	1274	1401	1680	2856	3781	4643
B6	49	165	373	649	1030	1715	1984	2208	2711	3792	4603	− *
B7	76	252	570	895	1416	2280	2744	3094	4172	4495	−	−
B8	85	283	641	1137	1846	2745	3201	4048	4673	−	−	−
B9	96	321	715	1255	2071	3210	3861	4546	−	−	−	−
B10	100	333	757	1315	2163	3615	4325	−	−	−	−	−

* The sensor is ruptured by strain overloading.

**Table 16 sensors-19-01840-t016:** The strain on the bolts at each loading step in E8 (unit: MPa).

Bolt	y (mm)											
	1	2	3	4	5	6	7	8	9	10	11	12
B1	−2	−8	−18	−28	−61	−79	−130	−136	−283	−364	−476	−502
B2	0	−4	−10	−15	−32	−41	−71	−90	−232	−274	−376	−395
B3	0	−1	−2	−3	−7	−10	51	98	167	195	440	1099
B4	7	24	52	79	168	222	336	542	737	1096	1947	2355
B5	27	77	165	253	539	710	1097	1466	1649	2268	3114	3721
B6	41	114	245	375	801	1055	1735	2018	2701	3195	4025	4549
B7	57	158	340	521	1113	1465	2274	2795	3587	3794	4524	−
B8	76	209	448	687	1466	1930	3000	3675	4083	4524	−	−
B9	85	233	499	766	1635	2152	3347	4097	4564	−	−	−
B10	95	259	557	854	1823	2400	3623	4565	−	−	−	−

**Table 17 sensors-19-01840-t017:** Comparison between (*EI*)*^d^*/(*EI*)*^ud^* and *H*_FM1_ in E10.

	*y* (mm)							
	1	2	3	4	5	6	7	8
*M* (kN·m)	0.400	2.099	4.300	6.398	8.493	9.497	10.296	11.198
*κ* (10^−3^ m^−1^)	0.043	0.226	0.463	0.689	0.915	1.205	1.443	1.807
*EI* (10^3^ kN·m^2^)	9.291	9.289	9.288	9.286	9.282	7.881	7.135	6.197
(*EI*)*^d^*/(*EI*)*^ud^*	1.000	1.000	1.000	1.000	1.000	0.848	0.768	0.667
	*y* (mm)							
	9	10	11	12				
*M* (kN·m)	12.000	12.801	13.701	14.100				
*κ* (10^−3^ m^−1^)	2.340	3.295	4.966	8.930				
*EI* (10^3^ kN·m^2^)	5.128	3.885	2.759	1.579				
(*EI*)*^d^*/(*EI*)*^ud^*	0.552	0.418	0.297	0.170				
	*y* (mm)							
	1	2	3	4	5	6	7	8
*η^ud^*	0.144	0.144	0.144	0.144	0.144	0.144	0.144	0.144
*η^d^*	0.144	0.144	0.144	0.144	0.144	0.125	0.114	0.099
*H* _FM1_	1.000	1.000	1.000	1.000	1.000	0.867	0.788	0.687
	*y* (mm)							
	9	10	11	12				
*η^ud^*	0.144	0.144	0.144	0.144				
*η^d^*	0.082	0.063	0.046	0.027				
*H* _FM1_	0.572	0.438	0.316	0.184				
	*y* (mm)							
	1	2	3	4	5	6	7	8
Error (%)	0.0	0.0	0.0	0.0	0.0	2.2	2.6	3.0
	*y* (mm)							
	9	10	11	12				
Error (%)	3.6	4.8	6.4	8.2				

**Table 18 sensors-19-01840-t018:** Comparison between (*EI*)*^d^*/(*EI*)*^ud^* and *H*_FM1_ in E8.

	*y* (mm)							
	1	2	3	4	5	6	7	8
*M* (kN·m)	0.372	1.949	3.992	5.944	7.886	8.819	9.560	10.399
*κ* (10^−3^ m^−1^)	0.040	0.210	0.430	0.640	0.849	0.950	1.212	1.482
*EI* (10^3^ kN·m^2^)	9.290	9.288	9.291	9.287	9.285	9.283	7.887	7.017
(*EI*)*^d^*/(*EI*)*^ud^*	1.000	1.000	1.000	1.000	1.000	1.000	0.849	0.755
	*y* (mm)							
	9	10	11	12				
*M* (kN·m)	10.898	11.598	12.499	12.800				
*κ* (10^−3^ m^−1^)	1.885	2.586	4.125	6.972				
*EI* (10^3^ kN·m^2^)	5.910	4.597	3.084	1.878				
(*EI*)*^d^*/(*EI*)*^ud^*	0.636	0.495	0.332	0.202				
	*y* (mm)							
	1	2	3	4	5	6	7	8
*η^ud^*	0.144	0.144	0.144	0.144	0.144	0.144	0.144	0.144
*η^d^*	0.144	0.144	0.144	0.144	0.144	0.144	0.125	0.112
*H* _FM1_	1.000	1.000	1.000	1.000	1.000	1.000	0.871	0.777
	*y* (mm)							
	9	10	11	12				
*η^ud^*	0.144	0.144	0.144	0.144				
*η^d^*	0.095	0.075	0.051	0.032				
*H* _FM1_	0.660	0.519	0.353	0.220				
	*y* (mm)							
	1	2	3	4	5	6	7	8
Error (%)	0.0	0.0	0.0	0.0	0.0	0.0	2.6	2.9
	*y* (mm)							
	9	10	11	12				
Error (%)	3.8	4.8	6.3	8.9				

**Table 19 sensors-19-01840-t019:** Comparison between *n^d^* from different sources in E10.

		*y* (mm)											
		1	2	3	4	5	6	7	8	9	10	11	12
**Table 4**	*n^d^*	18	18	18	18	18	17	15	13	11	9	7	5
**From Monitoring**	*n^d^*	18	18	18	18	18	18	15	13	11	9	7	5

**Table 20 sensors-19-01840-t020:** Comparison between *n^d^* from different sources in E8.

		*y* (mm)											
		1	2	3	4	5	6	7	8	9	10	11	12
**Table 4**	*n^d^*	18	18	18	18	18	18	17	15	13	11	9	7
**From Monitoring**	*n^d^*	18	18	18	18	18	18	17	15	13	11	9	7

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
