# Peer review of "Damage Detection and Evaluation for an In-Service Shield Tunnel Based on the Monitored Increment of Neutral Axis Depth Using Long-Gauge Fiber Bragg Grating Sensors"

_sensors, 2019, doi:10.3390/s19081840_

Reviewer 1 Report

This paper developed damgae detection technique by obtaining the damage index, D [equation (7)], based on FBG strain measurement such as the monitoring increment of neutral axis depth using long-gauge fiber Bragg sensors. Thus the paper demonstrated the good damage techniques. Nonetheless, it may be suggested that the followings are revised. (1) The paper describes sufficient background and provide appropriate research design and methoths. However, it is suggested that the results may be described in a few papes of manuscripts with core parts, e.g., within 4 or 5 pages, although the results clearly presented. It may be an example that the conclusion seems to be very concise. This may be scientific research article not the technical research report. (2) What is the applied load p (105 kN/m2), is it the force or pressure/stress? If it is the force whay not N, if it is pressure or stress, it should be in unit Pa? (3) Dimension of sensor package and that of ring would be more clearly depicted in the schematic. Thanks.

Author Response

Response to Reviewer 1 Comments

At first, we want to appreciate these useful suggestions from the reviewers. Based on their comments, we have revised our manuscript in detail. And English of this paper has been improved to further show the theoretical and practical applicability of the proposed damage detection and evaluation for in-service shield tunnel based on the monitored increment of neutral axis depth using long-gauge Fiber Bragg Grating(LFBG) sensors.

Point 1: The paper describes sufficient background and provides appropriate research design and methods. However, it is suggested that the results may be described in a few papes of manuscripts with core parts, e.g., within 4 or 5 pages, although the results clearly presented.

Response 1: Thank you for your suggestion.

(1)     In fact, the core part of this paper is the proposed damage index D [Equation (7)]. The whole derivation process is shown from P4 to P9. In Section 2.2(P5), we propose Eq.(7) to show the relationship between the damage index D and the ratio of the damaged EI and the non-destructive EI from Eq.(1)~Eq.(3). In addition, Eq.(7) indicates that D can be calculated based on the ratio of the number of the yield bolts and the sum of the bolts. But how do we calculate the number of the yield bolts? We notice that the bolt with a lower position than the neutral axis is likely to yield if the damage occurs, and the damage occurrence can be pointed out uniquely by the movement of the neutral axis(Section 3.1, P6). Thus the key problem is to measure the position of the neutral axis. We use the LFBG array shown in Fig.6(b), P8 to obtain the strain distribution in the vertical direction. The position of the zero strain is the position of the neutral axis. Therefore, if we get the position of the neutral axis, we can calculate two angles of ϕ and ω based on the plane-section assumption. The former represents the positon of neutral axis. The latter indicates the positon of yield line of the bolts. So, the number of the yield bolts can be obtained. Eq.(8)~Eq.(12) can be used to calculate the coefficient of the entire blots ηud and nud, and Eq.(13)~Eq.(18) can be used to calculate the coefficient of the yield blots ηd and nd. Then the value of D can be calculated by the obtained ηud, nud, ηd and nd.

Point 2: It may be an example that the conclusion seems to be very concise. This may be scientific research article not the technical research report.

Response 2: Thank you for your suggestion.

The conclusions have been revised as follows.

(1)     Based on the analysis of two deformation modes (i.e., bending mode and dislocation mode), three kinds of failure modes are assumed, including the bending failure mode, the shearing failure mode due to dislocation, and the shearing failure mode of the equivalent shear stiffness reduction, which resulted from the bolts subjected to an excessive bending moment. These three failure modes and their development were successfully detected and evaluated by the proposed damage index D.

(2)     The damage index D can be determined only by the monitored NAD, which is obtained from long-gage strain distribution of the LFBG sensors array. The calculation of damage index value is independent of the loading mode and the mechanical properties of materials.

(3)     The damage index D characterizes the development of damage in the tunnel accurately through calculating coefficients HFM1 and HFM3 in most cases. The maximum deviation between the HFM1 and the true decrease of the equivalent flexural stiffness is less than 1.7% in the numerical simulation result and less than 9% in the experimental result. The calculated HFM3  matches well with its true value both in the undamaged stage and the early-damaged stage in the numerical simulation and the experimental results. The accuracy of the index D shows great applicability of the proposed damaged detection and evaluation method in the practical monitoring and maintenance for in-service shield tunnels.

Point 3:  What is the applied load p (105 kN/m2)? Is it the force or pressure/stress? If it is the force whay not N, if it is pressure or stress, it should be in unit Pa?

Response 3: Thank you for your suggestion. p is the applied pressure in the model.

All kN/m2 have been revised to 103 Pa.

Point 4:  Dimension of sensor package and that of ring would be more clearly depicted in the schematic.

Response 4: Thank you for your suggestion. The diameter of each package of the LFBG sensor has been shown in Fig.4, P7.  The length of the sensor has been added in the last paragraph in P6 as “And the gauge length of the LFBG sensor can range from several centimeters to several meters”.

Figure 4.  The structural design of the packaged LFBG sensor

The dimension of the ring used in numerical simulation is shown in P9 as

The thickness of the segment is 0.3 m, and the width of the ring is 1.5 m. The outside diameter of the ring is 6.0 m and the inside 5.4 m.“

The dimension of the ring used in the experiment is shown in P15 as The outside diameter is 860 mm and the inside 780 mm. The width and the thickness of a segment ring is respectively 200 mm and 40 mm.

In addition, the English language and style of this manuscript has been carefully checked and revised. For example:

P9, the 2nd paragraph

Original manuscript:

The thickness of the segment is 0.3 m and the width of the ring is 1.5 m. The outside and inside diameters of the ring are 6.0 m and 5.4 m, respectively.

Revised manuscript:

The thickness of the segment is 0.3 m, and the width of the ring is 1.5 m. The outside diameter of the ring is 6.0 m and the inside 5.4 m.

P15, the 1st paragraph

Original manuscript:

The outside and inside diameters are 860 mm and 780 mm, the width of a segment ring is 200 mm, and the thickness of a segment is 40 mm.

Revised manuscript:

The outside diameter is 860 mm and the inside 780 mm. The width and the thickness of a segment ring is respectively 200 mm and 40 mm.

Reviewer 2 Report

The questions are:  Why suggest Bragg Long Range Networks ?  It seems to me that fiber optic sensors  of the Rayleigh type can very easily give  average deformation values over the entire  dimension to be monitored ? I wish I had an answer ...  Another issue is the linear placement of sensors.  This type of placement only makes it possible to obtain  axial deformations, whereas sinusoidal placement can provide  multi-parameter strain sensing ... It seems to me that distortions in the  other directions are also to be monitored.  What do the authors think ?  Response paths are suggested in the following 2 articles 1/ https://www.mdpi.com/1424-8220/17/4/667 2/ https://www.mdpi.com/1424-8220/19/3/742

Author Response

Response to Reviewer 2 Comments

At first, we would like to express our appreciation to these useful suggestions from the reviewers. Based on their comments, we have revised our manuscript in detail. The English writing of this paper has been improved to further show the theoretical and practical applicability of the proposed damage detection and evaluation for in-service shield tunnel.

Point 1: Why suggest Bragg Long Range Networks? It seems to me that fiber optic sensors of the Rayleigh type can very easily give average deformation values over the entire dimension to be monitored ? I wish I had an answer ...

Response 1: Thank you for your suggestion.

In theory, the strain distribution in tunnel can be measured by several fiber optic sensors. However, there are some restrictions in practical monitoring. At first, several sensors in a ring can be connected by a common fiber optic to reduce the difficulty of placement or measurement. Then the broken sensor must be replaced easily. The third one is that the sensor placed on the surface of the ring must be small, thin and slender enough to reduce the influence on other instruments. The last one is the price of the sensor must be cheap enough.

We compare many kinds of fiber optic sensors in China. The LFBG sensor may be the most suitable choice.

Point 2: Another issue is the linear placement of sensors. This type of placement only makes it possible to obtain axial deformations, whereas sinusoidal placement can provide multi-parameter strain sensing ... It seems to me that distortions in the other directions are also to be monitored. What do the authors think? Response paths are suggested in the following 2 articles.

1/ https://www.mdpi.com/1424-8220/17/4/667 2/ https://www.mdpi.com/1424-

8220/19/3/742

Response 2: Thank you for your suggestion.

         In Section 2.1(P3), this paper points out that there are only three kinds of damage modes in the in-service tunnel. However, the paper also indicates that only the bending mode and the shear failure mode of equivalent shear stiffness reduction caused by an excessive bending moment are possible to detect and evaluate in practice. In fact, these two modes belong to the bending failure. The linear placement of sensors is suitable for the detection of bending failure.

Figure 1  Two neutral   axises in a 3D tunnel

   However, the tunnel is a 3D structure. Thus in actual monitoring, we must place sensors shown as Fig.1 to detect the bending damage in two directions. Furthermore, we use a similar method to monitor the 3D movement of the tunnel. This paper is attached in our response letter.

[1]      Shen Sheng, Wu Zhishen, Lin Meijin. Distributed Settlement and Lateral Displacement Monitoring for Shield Tunnel Based on an Improved Conjugated Beam Method[J]. ADVANCES IN STRUCTURAL ENGINEERING,16(8),1411-1425.(SCI、EI)

In addition, the English language and style of this manuscript has been carefully checked and revised. For example:

P9, the 2nd paragraph

Original manuscript:

The thickness of the segment is 0.3 m and the width of the ring is 1.5 m. The outside and inside diameters of the ring are 6.0 m and 5.4 m, respectively.

Revised manuscript:

The thickness of the segment is 0.3 m, and the width of the ring is 1.5 m. The outside diameter of the ring is 6.0 m and the inside 5.4 m.

P15, the 1st paragraph

Original manuscript:

The outside and inside diameters are 860 mm and 780 mm, the width of a segment ring is 200 mm, and the thickness of a segment is 40 mm.

Revised manuscript:

The outside diameter is 860 mm and the inside 780 mm. The width and the thickness of a segment ring is respectively 200 mm and 40 mm.

   P22  The conclusion

Original manuscript:

(1)     The NAD is sensitive to the damage occurrence in a shield tunnel as shown by the numerical simulation and the scaled-down experiment.

(2)     The NAD and the damage index D are determined only by the long-gage strain measurements and the height coordinates of the sensors typical for daily monitoring. The calculations are independent of the loading mode and the mechanical properties of materials.

(3)     In most cases of ductile damage, the proposed damage index D is mainly composed of the ratios HFM1 and HFM3. The index is applicable to evaluate the damage levels for three different kinds of failure modes. Results showed that the maximum deviation between the HFM1 and the true decrease of the equivalent flexural stiffness is less than 1.7% in the numerical simulation, and less than 9% in the experiment. The HFM3 is accurate in the undamaged stage and the early-damaged stage both in the numerical simulation and the experiment.

Revised manuscript:

(1)     Based on the analysis of two deformation modes (i.e., bending mode and dislocation mode), three kinds of failure modes are assumed, including the bending failure mode, the shearing failure mode due to dislocation, and the shearing failure mode of the equivalent shear stiffness reduction, which resulted from the bolts subjected to an excessive bending moment. These three failure modes and their development were successfully detected and evaluated by the proposed damage index D.

(2)     The damage index D can be determined only by the monitored NAD, which is obtained from long-gage strain distribution of the LFBG sensors array. The calculation of damage index value is independent of the loading mode and the mechanical properties of materials.

(3)     The damage index D characterizes the development of damage in the tunnel accurately through calculating coefficients HFM1 and HFM3 in most cases. The maximum deviation between the HFM1 and the true decrease of the equivalent flexural stiffness is less than 1.7% in the numerical simulation result and less than 9% in the experimental result. The calculated HFM3  matches well with its true value both in the undamaged stage and the early-damaged stage in the numerical simulation and the experimental results. The accuracy of the index D shows great applicability of the proposed damaged detection and evaluation method in the practical monitoring and maintenance for in-service shield tunnels.

 Round  2

Reviewer 1 Report

Suggesed to change

1) all the unit of 103Pa to kPa for the entire manuscript.

2) for example, Es=3.45×104 MPa to 34,5 GPa, and Eb=2.06×105 MPa to 206 GPa (and Es=3.5×104 MPa, Eb=1.90×105 MPa).

Thanks!

Author Response

Response to Reviewer 1 Comments

At first, we would like to express our appreciation to these useful suggestions from the reviewers. Based on their comments, we have revised our manuscript in detail. The English writing of this paper has been improved to further show the theoretical and practical applicability of the proposed damage detection and evaluation for in-service shield tunnel.

Point 1: All the unit of 103Pa to kPa for the entire manuscript

Response 1: Thank you for your suggestion. All 103Pa have been revised to kPa. 

Such as the paragraph at P10,line 338~355

Table 1 and Table 2 show the tensile and compressive strain measurements for E9 and E6 at each loading step. The tensile strain is defined to be positive. Figure 10 shows that there are approximately linear relationships between the measured strains and the heights of the sensors. Thus, the method of least squares easily obtains the linear equations of kε+c for each linear relationship. These equations are used to construct Table 3 that shows the coefficient of determination R2, and the parameter c that is equal to the monitored NAD χ for each equation. The NAD of E9 remains approximately 2.560 m when the applied load p is less than 105kPa. The value of χ begins to increase when p reaches 105kPa. For E6, the same trend of χ also occurs for p greater than 120kPa. Thus, the proposed detecting method in Section 3 defines = 105kPa and = 120kPa as the threshold values of structural damage in E9 and E6. The stresses on bolts at different loading steps for E9 and E6 are listed in Table 4 and Table 5, respectively. Table 4 shows that the stress of the outermost bolt B9 in the tensile region of E9 reaches 661 MPa for a p of 105kPa. This stress value of this bolt slightly exceeds the yield stress of 640 MPa. Table 5 shows that the stress of B9 in the tensile region of E6 reached 651 MPa at a p of 120kPa. This bolt stress is also larger than the 640 MPa yield stress. These stress values indicate that the NAD χ is a sensitive index that can indicate the occurrence of damage. Moreover, the applicability of the plane-section assumption is verified since the coefficient of determination R2 of each best-fitting straight line is greater than 0.997 for each loading step before and after damage.

Point 2: for example, Es=3.45×104 MPa to 34,5 GPa, and Eb=2.06×105 MPa to 206 GPa (and Es=3.5×104 MPa, Eb=1.90×105 MPa).

Response 2: Thank you for your suggestion.

P9, line300

The numerical segment ring built in the ANSYS software is based on the real segment ring in the Shanghai Metro Line No. 1. The mechanical properties of the segment and bolt are as follows: Es=34.5GPa, Eb=206GPa, μs=0.167, μb=0.2, the compressive strength of the concrete =32.4MPa, the tensile strength of the concrete=1.89MPa, the yield strength of a bolt= 640 MPa(the yield strain of a bolt=3107με, correspondingly), the ultimate strength of a bolt=800 MPa, and kj2/kj1=0.1.

P15, line407

The mechanical properties of the segment and bolt are obtained by laboratory tests and are as follows: Es=35GPa, Eb=190GPa, μs=0.2, μb=0.3, the compressive strength of the concrete=41.4 MPa, the yield stress of a bolt=710.08 MPa, the yield strain of a bolt=3680 με, kj2/kj1=0.001.

Reviewer 2 Report

Dear Authors,

I still have comments

Many points should be considered.

The first one, it seems to me that to increase the accuracy of measurements, there is a strong need for at least 2 to 3 fibers to be placed at the area where strain must be monitored. You must keep in mind that shrinkage of concrete on the fiber, local compression on the Bragg and other issues can make bring signicant uncertainties to the measurements ...

The 2nd point is that even with long range fiber-Bragg sensors the strain will be only measured on a small surface ... Will this be representative of strain magnitude over the whole area that has to be monitored ...

The 3rd point is related to multiparameter strain monitoring ... This should be at least mentioned in the paper ... One article dealing with this issue has been already suggested to be considered by authors and added to the reference list. Indeed, one option is to embed 2 types of sensors ... Used ones along with the use of Raighleigh sensors that will enable comparison to be done with Braggs meanwhile monitoring the whole area ... In addition, if the placement is sinusoidal, not only you will have access to tensile strain, but also to the other components of strain tensor, including bending ... If in the discussion part, what is mentioned above is emphasized, it will enhance the quality of the article, thus providing stronger interest to readers. What is asked to the authors is not necessarely to make additional experiments and associated nurmerical simulations, but to consider and then to mention from prospective point of view that additional options are available to provide more accurate research work in the future, for the benefice of research on smart structures.

Author Response

Response to Reviewer 2 Comments

At first, we would like to express our appreciation to these useful suggestions from the reviewers. Based on their comments, we have revised our manuscript in detail. The English writing of this paper has been improved to further show the theoretical and practical applicability of the proposed damage detection and evaluation for in-service shield tunnel.

Point 1: The first one, it seems to me that to increase the accuracy of measurements, there is a strong need for at least 2 to 3 fibers to be placed at the area where strain must be monitored. You must keep in mind that shrinkage of concrete on the fiber, local compression on the Bragg and other issues can make bring signicant uncertainties to the measurements ...

Response 1: Thank you for your suggestion.

This sentence has been added at P7, Line 251.

In addition, it may need to place a few LFBG sensors, sometimes associated with some other distributed sensing technique[53,54] in one monitoring position in practice to not only ensure the accurate measurements for a long time but also eliminate the effect of some unpredicted factors such as shrinkage of concrete on the fiber, local compression on the Bragg and so on.

1.          Drissi-Habti, M.; Raman, V.; Khadour, A.; Timorian, S.  Fiber Optic Sensor Embedment Study for Multi-Parameter Strain Sensing. Sensor 2017, 17,667. doi:10.3390/s17040667

2.          Raman, V.; Drissi-Habti, M.; Limje, P.; Khadour, A. Finer SHM-Coverage of Inter-Plies and Bondings in Smart Composite by Dual Sinusoidal Placed Distributed Optical Fiber Sensors. Sensors 2019, 19, 742; doi:10.3390/s19030742

Point 2: The 2nd point is that even with long range fiber-Bragg sensors the strain will be only measured on a small surface ... Will this be representative of strain magnitude over the whole area that has to be monitored ...

Response 2: Thank you for your suggestion.

         The gauge length of a LFBG sensor can range from several centimeters to several meters. In the longitudinal direction, the length of the influencing area of the joint is about 750px~1000px, which equals to (0.2~0.3)* the length of the whole ring. So the LFBG sensor can measure the average strain of an area with a length of 750px~1000px easily.

         As shown in Figure 1, the array of sensor is also used to determine the best-fit of the straight line y = kε+c based on the height coordinate of S1–Sn and the corresponding strain measurements ε1–εn. The fitted result can represent the linear strain distribution in the influencing area of the joint in the vertical direction.

(a)     Circumferential section

(b)       Longitudinal   section

(c) Fitting line based on LFBG measurements

Figure 1. Determination of the NAD in a segment ring based on LFBG strain measurements

Point 3: The 3rd point is related to multiparameter strain monitoring ... This should be at least mentioned in the paper ... One article dealing with this issue has been already suggested to be considered by authors and added to the reference list. Indeed, one option is to embed 2 types of sensors ... Used ones along with the use of Raighleigh sensors that will enable comparison to be done with Braggs meanwhile monitoring the whole area ... In addition, if the placement is sinusoidal, not only you will have access to tensile strain, but also to the other components of strain tensor, including bending ... If in the discussion part, what is mentioned above is emphasized, it will enhance the quality of the article, thus providing stronger interest to readers. What is asked to the authors is not necessarely to make additional experiments and associated nurmerical simulations, but to consider and then to mention from prospective point of view that additional options are available to provide more accurate research work in the future, for the benefice of research on smart structures.

Response 3: Thank you for your suggestion.

This sentence has been added at P7, Line 251.

In addition, it may need to place a few LFBG sensors, sometimes associated with some other distributed sensing technique[53,54] in one monitoring position in practice to not only ensure the accurate measurements for a long time but also eliminate the effect of some unpredicted factors such as shrinkage of concrete on the fiber, local compression on the Bragg and so on.

3.          Drissi-Habti, M.; Raman, V.; Khadour, A.; Timorian, S.  Fiber Optic Sensor Embedment Study for Multi-Parameter Strain Sensing. Sensor 2017, 17,667. doi:10.3390/s17040667

4.          Raman, V.; Drissi-Habti, M.; Limje, P.; Khadour, A. Finer SHM-Coverage of Inter-Plies and Bondings in Smart Composite by Dual Sinusoidal Placed Distributed Optical Fiber Sensors. Sensors 2019, 19, 742; doi:10.3390/s19030742

Round  3

Reviewer 2 Report

The article is fine